# Parental experience of the diagnostic process and its role in the decision to terminate pregnancy due to fetal abnormality; A qualitative interview study

**Maiken Fabricius Damm**[1,2,3], **Dorte Hvidtjoern**[1,4], **Puk Sandager**[1,4,5], **Ida Vogel**[1,4,5], **Christina Prinds**[2,3,6], **Stina Lou**[4,5,7]*

1 Department of Obstetrics and Gynecology, Aarhus University Hospital, Aarhus, Denmark, 2 Department of Women's Health, University Hospital of Southern Denmark, Odense, Denmark, 3 Department of Regional Health Research, University of Southern Denmark, Odense, Denmark, 4 Department of Clinical Medicine, Aarhus University, Aarhus, Denmark, 5 Center for Fetal Diagnostics, Aarhus University Hospital, Aarhus, Denmark, 6 Department of Clinical Research, Unit of Gynecology and Obstetrics, Odense University Hospital, Odense, Denmark, 7 DEFACTUM – Public Health Research, Central Denmark Region, Aarhus, Denmark

* Stina.lou@clin.au.dk

## Abstract

### Introduction

The detection of fetal malformation is a shock to expectant parents and often initiates a diagnostic process of additional tests and ultrasound scans, that may be uncertain and stressful to the parents. The aim of the present study was to investigate how expectant parents experienced the diagnostic process and how their decision to terminate the pregnancy was reached during that process.

### Methods

Semi-structured interviews with 11 Danish women and nine male partners one to five months after termination of pregnancy. All interviews were conducted in the participants' homes and lasted 97–135 minutes. Thematic analysis was performed.

### Results

1) The theme, '*In no man's land'*, describes the two-phased diagnostic process: First, the initial shock of a potential ultrasound finding, and the uncertain – but still hopeful – days of waiting for a follow-up scan and specialist consultation. Second, the feeling of professionalism and companionship when interacting with the fetal medicine specialists, who still could not always provide the hoped-for answers. 2) The theme, '*Inescapable decision*', describes how decision-making oscillated as new information or potential interpretations entered the diagnostic process. The participants described a continuous contemplation of the inevitable final choice regarding continuation or

**Data availability statement:** The data underlying this study consist of in-depth qualitative interviews containing potentially identifiable and sensitive personal information. Due to the highly individual nature of the material, full transcripts cannot be fully anonymized, and public sharing would not comply with the ethical approval granted for this study. Participants consented only to external data sharing in anonymized form. Data access requests may be directed to DEFACTUM, Central Denmark Region, at defactum@rm.dk, and will be considered for researchers who meet the criteria for access to confidential data in accordance with the Central Denmark Region's data protection policies.

**Funding:** This work was funded by University Hospital of Southern Denmark (grant#23) and by Aarhus University Hospital, (grant#SVP101121 and SVP040723).The funders had no role in study design, data collection and analysis, decision to publish, or preparation of the manuscript.

**Competing interests:** The authors have declared that no competing interests exist.

termination. Being in this process – for days or weeks – was described as an emotional rollercoaster with feelings of both hope and despair until the final decision to terminate was made.

## Conclusion

During a prenatal diagnostic process parents must endure uncertainty, waiting times and an ongoing oscillation between hope and no-hope for the pregnancy. However, the diagnostic process may also be understood as an opportunity for dialogue, reflection and adjustment, allowing for a personal and well-considered decision, even if painful.

## Introduction

When fetal disease/abnormality is detected by prenatal ultrasound, appropriate and empathetic communication with expectant parents is paramount [1–5]. Some ultrasound findings show severe disease/abnormalities, while others have a more unpredictable or variable prognosis [6,7]. Regardless of the assessed severity, an abnormal prenatal finding is a shock to expectant parents [3,5,8–11], as the health of the fetus and the hoped-for future are questioned [5,11]. An ultrasound finding may lead to a diagnostic process characterized by waiting (e.g., for the fetus to grow, for test results to come back) and by limited availability of prognostic information as the extent, development and potential implications of a detected abnormality/disease can be difficult to predict. All diagnostic trajectories are distinctive and highly individual, but all take an emotional toll on parents as new information (or lack thereof) is assessed, and potentially complex decision-making about the continuation or termination of pregnancy must be made [3,9,12]. Women's *reasons* for terminating a pregnancy with severe fetal malformation/disease are well-described and include concern for the baby, potential siblings and the family as a whole [12,13]. However, less is known about the *process* of reaching this decision. Such insights are essential for healthcare professionals aiming to provide patient-centered care and support expectant parents in making well-informed and value-consistent reproductive choices.

Thus, the aim of this study was to investigate expectant parents' lived experiences of the diagnostic process, from suspicion of fetal abnormalities to the final decision to terminate pregnancy in the second trimester.

## Methods

A qualitative interview study was chosen to answer the research question. The analysis followed a phenomenological-hermeneutic approach inspired by Gadamer [14], viewing understanding as a dialogical and interpretive process. Phenomenology guided attention to participants' lived experiences, while the hermeneutic lens enabled interpretation through a reflexive engagement with the data. Meaning was developed through a dynamic movement between parts and whole, acknowledging the influence of the researchers' pre-understandings.

## Study setting

In Denmark, all pregnant women are offered full-coverage, tax-financed antenatal care, including a first and second-trimester ultrasound examination. Sonographers (nurses/midwives with Fetal Medicine Foundation certification) routinely perform and communicate these scans to expectant parents. If abnormalities are suspected, the woman is referred to a fetal medicine specialist for follow-up, typically within one to five days. At Aarhus University Hospital, where the study was conducted, the Unit of Fetal Medicine manages the complete diagnostic process including follow-up examinations, specialist evaluations and genetic testing and diagnostics. Women undergoing termination of pregnancy due to fetal abnormality (TOPFA) are managed by specialised midwives at the Unit of Perinatal Loss [15,16].

In Denmark, abortion is a legal right until gestational week 12 [17]. After gestational week 12, TOPFA must be approved by a regional specialist council, but in cases of severe fetal anomaly or disease, TOPFA is rarely denied [17,18].

## Recruitment and participants

Eligible participants were recruited from the Unit of Perinatal Loss. Inclusion criteria were TOPFA from gestational week 17 and ability to be interviewed in Danish or English. Eligible participants received written and verbal study information before consenting to be contacted by the first author within 2–8 weeks. A total of 35 eligible participants consented; however, 15 later withdrew due to lack of energy/time. Consequently, 11 women and nine male partners were interviewed. Sample characteristics are presented in Table 1.

## Ethical statement

Interviews were performed following the regulations of the Code of Ethics of the World Medical Association [19]. The study was reviewed by the National Committee on Health Research Ethics (J.no: 210521–89/2021) and registered with the Internal Research Register in the Region of Central Denmark (J.no 1-16-02-198-21). Participants received written and verbal information about the study on several occasions and gave written consent before the interviews. Consent could be withdrawn at any time.

## Analysis

The interviews were based on a semi-structured interview guide [20]. The interview guide was developed by the interdisciplinary research team and based on current scientific literature and their extensive clinical and research experience. The interviews were performed by the first author between June 2021 and August 2022, one to five months after termination of the pregnancy. The timing of the interview was based on participant' preferences as some participants did not feel ready for an interview when initially approached (2–8 weeks after termination). The interviewing approach was open-ended and flexible, with an aim to provide opportunity to freely express any understandings and experiences important to the individual participants. The semi-structured interview guide served to support participants' recollections while ensuring that the research question was fully explored. By participant choice, interviews were conducted in their homes, lasting 97–135 minutes. All interviews were digitally recorded, transcribed verbatim, and discussed in the research group as part of the

**Table 1. Sample Characteristics.**

| Primiparous | 6 |
|---|---|
| Multiparous | 5 |
| In a relationship | 10 |
| Single | 1 |
| Age of participants (range) | 27–42 |
| Gestational age at TOPFA (weeks+days) | 17 + 3–22 + 2 |

analytical process and researcher triangulation. Recruitment continued until sufficient information power was estimated to have been met [21].

Data were analyzed using reflexive thematic analysis as formulated by Maguire & Delahunt [22]. Focusing on data relevant to the research question, five initial themes, capturing the essence of data, were developed by the first author and discussed with the last author. The first author then coded data into five themes, resulting in five data items. All items were re-read, and chunks of data relevant to the research question were coded, in a process of redefining and refining the themes. At this point, the preliminary analysis and interpretation were discussed among all authors and two overarching themes were identified. Fig 1 presents the analytical process.

## Results

The results centered on two themes: 1) the diagnostic process with repeated scans, further testing and waiting time, described as being "*In no man's land*", and 2) the process of reflections that shaped decision-making, described as an "*Inescapable choice*" (See Table 2 for an overview of the themes).

In total, the analysis unfolded the experiences of an emotional and surreal rollercoaster ride of hope/no hope, frustration, and grief alongside existential reflections, ambivalent feelings, and bodily sensations. For the participants, the ride felt very steep at the beginning of the diagnostic phase, had various bumps along the way and peaked again before the decision to terminate was settled. This process is described in more detail below.

### Theme 1: In no man´s land

This theme described the participant's two-phased experiences of first, reactions to abnormal prenatal findings, and second, the experience of waiting and assembling pieces of information and test results, of dialogues and reflections.

**A shocking turn of events.** For most participants, the first suspicion of fetal abnormality was detected at the second-trimester ultrasound scan. At different point during the scan, all participants experienced a change in the room's atmosphere, from positive to tense. Participants sensed the sonographer's silence, focused attention, and quiet dialogue with colleagues.

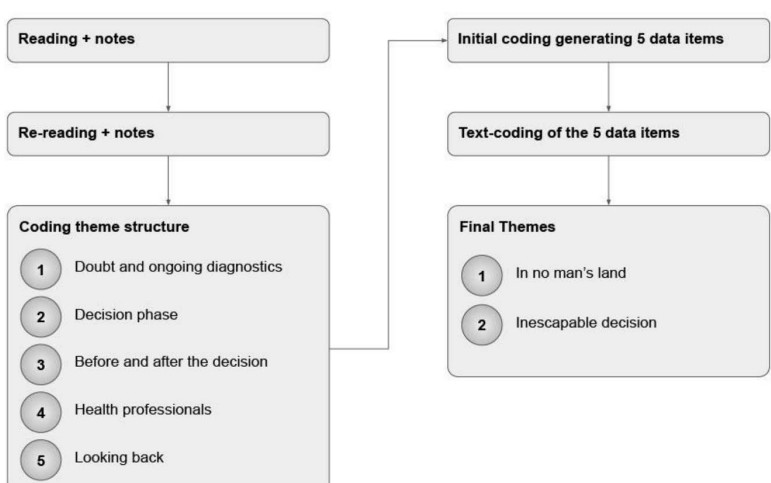

**Fig 1. Analytical process.**

**Table 2. Comparative thematic analysis summary.**

| Analytical Dimension | Theme 1: 'In no man's land' | Theme 2: 'Inescapable decision' |
|---|---|---|
| **Temporal focus** | Early to mid-diagnostic phase | Mid to late diagnostic phase, decision-making point |
| **Dominant emotional tone** | Shock, hope, confusion, isolation | Ambivalence, sorrow, clarity, responsibility |
| **Main parental needs** | Information, clarity, reassurance | Time, moral certainty, alignment between partners |
| **Healthcare professional role** | Gatekeeper, then companion; source of both frustration and trust | Supportive validator; respected but unable to make the decision |
| **Role of time** | Perceived as unbearable, dragging, yet necessary for information accumulation | Experienced as preparatory, yet ambivalent and emotionally burdensome |
| **Embodiment** | Emergent awareness of bodily connection to fetus | Heightened bodily experience; fetal movement as source of doubt and bonding |
| **Decision orientation** | Open-ended, filled with what-ifs and searching for best-case scenarios | Focused on potential suffering and family impact; emotional rationalization of a painful choice |

*"Mum: And then the atmosphere just changed completely, it was just really obvious that something was not right. And the doctor called a colleague, and well, […] … you could just tell by the look on their faces that they were kind of like… looked at each other, whispering a bit and then…" (interview 10)*

Some women explained that things happened "in slow motion" and that the information from the sonographer was vague but worrying. The information that a follow-up scan by a fetal medicine specialist was necessary was described as a shock, initiating a long list of questions that, with much frustration and concern, could not be answered by the sonographer.

*"Dad: What we needed was a clear indication of where this was going. It was obvious… from their faces, that this wasn't good. It just wasn't good. It would have been nice if they had said that as well" (interview 9)*

All participants recollected the wait for the follow-up examination as mentally and emotionally challenging. They suspected that the sonographer had withheld information and that something was wrong, but they also described feeling glimpses of hope. They were desperate for information and felt isolated with *'loads of questions'*, but could do nothing but wait:

*"Dad: I think, right at the beginning, we were sort of in shock […] …because we didn't know where we were going and… there were just worst-case-scenarios running through our minds, you think the worst. Yes. So, I remember it as some really emotional days, we cried a lot." (interview 7)*

A few women experienced an immediate negative reaction and '*felt troubled*' by carrying a potentially sick fetus/baby. Others described feeling a stronger bond with the baby after the suspicion was raised; feeling protective and *'clinging on to the hope'* that the suspected seriousness of the finding would later be disregarded as a mistake or simply the result of poor sonographer skills.

*"Mum: Well, we knew it wasn't good but… I don't know… we still had this tiny hope that they had misread the scan or that the equipment was broken or…but it was just like, it was really crazy those days" (interview 10)*

**Waiting for answers.** The awaited appointment with the fetal medicine specialist was generally described positively. They had informed the participants that they would first scan and then inform, which produced a feeling of being involved and acknowledged rather than 'kept in the dark'.

*"Dad: She said"I'll do the scan and then we'll talk". […] And afterwards she asked about what the sonographer had told us […] And then she said that she was very worried […] She was just so professional, and it is really hard to deliver such bad news…" (interview 3)*

From this point forward, the participants generally described healthcare professionals as thorough, persistent, and empathic, contributing to the participants' trust in the accuracy and interpretation of findings/test results. In the subsequent trajectory, the healthcare professionals acted forthcoming and flexible with the process well-adjusted to individual needs, resulting in feelings of humanized care.

*"Mum: So, I asked them for these extra scans... could we just keep an eye on [the fetus/baby]? […] I really needed it to be like; "now it's over, now it's for sure". So, they gave me the scans and at that last one she [the fetal medicine specialist] said; "there are still no changes… [the fetal malformation] really is…". And then I was in a mental condition to ask "well okay, now I'm ready, what is going to happen next?"" (interview 6)*

Waiting time was nerve-wracking for all participants, though the actual time-in-waiting differed significantly depending on the diagnostic tests needed. Also, holidays, staff shortages, and test turnaround times prolonged waiting time, which was frustrating.

*"Mum: It was a while until I got the answer (from the CVS). It was like a week... Then I was contacted that they had found something in the CVS and wanted to do an amniotic test. And then I had to wait again for that and then again wait for the answer. It's just a long wait…" (interview 2)*

At this stage, the rollercoaster of feelings fluctuated between hope and no hope, between grief and 'what if's': What if the condition could be tolerable for the child? What if the condition could be healed? The state of hope/no hope was described more for participants whose prenatal findings had initially appeared less severe or more uncertain. Waiting time was described as a demanding process as information accumulated, and participants reflected for and against termination, considering test results and weighing up the input from specialist consultations.

*"Mum: At that point I was like; "of course I will have my baby, he's doing fine.." But then we read about it and noticed this strange face they had (people with same condition as fetus), and we know how awful children can be towards each other, and grown-ups as well, if you are different and… We're that desperate that we immediately order a 3D scan […] if he has a normal-ish face, then it's okay. Then we will have him for sure. But we read even more, about psychiatric symptoms […] and the physical part […] We just read a lot and were just insanely confused" (interview 11)*

Adding to the emotionally demanding process was the challenges of understanding information about rare diseases, statistical probabilities and prognostic uncertainty. The healthcare professionals admitted that information was often less detailed and precise than desired, underlining genuine feelings of inadequacy from healthcare providers and expectant parents. There was parental frustration when specialists used diffuse and generic terms such as "a child with special needs", as it was too imprecise. However, the participants generally appreciated how healthcare professionals' honesty, proficiency and care helped ground them, shouldering the experience in some sort of companionship.

*"Mum: But the doctor was just really calm about it. When she had explained things, she was quiet for a few minutes… letting us adapt and ask questions. And well, she just seemed so honest. […] Dad: She was genuinely sad on our behalf. You could tell that it affected her too, having to deliver such bad news. […] It wasn't cold and cynical, no. So that helped a lot"* (interview 3)

Most participants searched for additional information on many different platforms throughout the diagnostic process, hoping to find answers. While acknowledging the irrational aspect, some participants described searching for positive stories of mistaken diagnoses or better prognoses, as there was a feeling that meetings with the specialists rarely included anything positive.

*"Dad: I think we looked for some expert to say "hell yes, you can do that! It's no trouble at all". I think that's what we needed … but no one ever said that…"* (interview 11)

As the pregnancy for all participants was advanced, all had involved family and friends in the news on their pregnancy. Involving and informing family and friends about this new and serious situation was a valued source of support, but also burdensome. It was particularly difficult because the information was uncertain, the prognosis changed, and more people had to be informed as the waiting time continued. Having to change/not attend planned social events, triggered feelings of being '*misplaced*' and '*not recognized,*' initiating a surreal feeling that the participants' world 'stood still' while the 'outside world' moved on as if nothing had happened.

*"Mum: And the world just kept spinning. The things in television – it was just; "seriously?! Are you aware of our situation!?" […] And Instagram and everybody's just doing so well, and we're just in the middle of this totally horrible… I think that was hard as well. That life goes on …"* (interview 6)

## Theme 2: Inescapable decision

This theme described the decision to terminate the pregnancy as an overwhelming and sometimes lengthy process involving emotion, information, and back-and-forth dialogue. The participants described the decision as very tough, yet – ultimately – 'relatively clear-cut' for most. The main concern was that the unborn child, the current/future siblings and the family as a whole would suffer. Several wished for 'a crystal ball image' of the child's potential future, but incoming information regarding the severity of the child's condition often made the decision less ambiguous.

*"Dad: well, for sure we wanted a crystal ball or …. – if we could see how it was going when she (the baby) was two or three years old but… Well, I think the doctor was really good saying; these are the odds… or, he used the expression that this is the forecast we've got now and it's just really bad news for her in regards to…. having a normal life or not being burdened by severe disabilities.*

*Mum: Of course, we wanted more information, but it just didn't exist…"* (interview 7)

Some couples experienced asynchronous decision-making, where the partner sometimes reached a final TOPFA conclusion '*faster*'. Some attributed this to the partner's lack of physical involvement, as the bodily distance and absence of physical sensations (as experienced by the pregnant partner) made them feel more detached from the unborn baby/fetus. Some fathers emphasized the importance of not forcing the decision-making of their pregnant partners, and many participants highlighted the importance and value of intra-couple agreement in the decision.

*"Dad: With all other decisions, I'm the guy that says; "now, this is how we do it." But with this I just knew, if you (partner) had said that… as I had my opinion already… But I could never force it on you, I couldn't do that. We needed to make this decision together" (interview 6)*

Some participants leaned on pre-pregnancy discussions of an expected preference for TOPFA in case of a severely affected child. However, facing realities rather than hypothetical choice made the participants understand and acknowledge the emotional and ethical complexities of the decision. Some participants asserted that this decision had to be rational, meaning that emotions should (or had to be) set aside.

*"Dad: We really tried to make a non-emotional decision. As in, okay, I know this is tough and it's emotional and everything, but the decision is forever. So, it has to be logical at first. […] And well, we succeeded in projecting and thinking case scenarios […] trying to put in as many elements as possible.*

*Mum: I think we agreed throughout the process and we both felt that we did the right thing and felt comfortable about it, even though it didn't make it easier…." (interview 5)*

However, concurrent disquiet seemed present in all participants despite their certainty and lack of decisional regret. They expressed ambivalence of making this life-changing decision on their own, with no one to tell them what to do. Therefore, some participants found the TOPFA approval to acknowledge their decision as acceptable and reasonable.

The time between approval and TOPFA procedure was described as ambivalent: termination was planned, but the baby/fetus was still alive and moving.

*"Mum: It was horrible (indicates baby moving) [cries] […] Then it's not just a couple of split cells … when it kicks inside the belly [emotional]. Then it's really starting to remind you of a baby. So... it was heartbreaking you know…" (interview 8)*

Many women described an inescapable bodily feeling of carrying the baby/fetus, which, for some, prolonged the decision-making process. Also, some women described deliberately *'trying to bond'* with the baby/fetus during these last days of carrying.

*"Mum: I remember I started talking to him because I thought; "this is what he gets. When he comes out, he will be dead [cries]. So It's kind of now I have to do this".*

*Interviewer: What did you tell him?*

*Mum: Sometimes it was just like everyday things. But I think I also told him that I was sorry. That this should happen to him" (interview 9)*

Some women described ambivalence as the fetal movements '*sparked joy*' and caused them to question the severity of fetal disease, which in turn '*disturbed their rationality*' and, for a few, even the certainty of their decision.

*"Mum: it was at the same time lovely [crying] and then it was hard as well because... I felt a bit like she wanted me to stop this. Like she was giving a sign that... I'm here and everything is normal" (interview 2).*

Meanwhile the fathers could opt in or out of connecting with the baby/fetus in the time between making the decision until giving birth and terminating the pregnancy.

*"Dad: this is what you're able to forget as a man, the things that are physical about it – it's easier to distance yourself from it…" (interview 6)*

Thus, the fathers were very aware of their pregnant partner's bodily involvement in the decision and that her '*burden*' was different to his. For example, one father explained his need to take part in the "carrying" the fetus/baby:

*"Dad: At the funeral service, I carried the coffin. I think it was kind of… Now E (his partner) carried him until he was born [cries] and then I carried him in the end. Then we sort of shared this…" (interview 4)*

The diagnostic process was shorter for some participants with severe prenatal findings, while others had much longer processes with serial scans and tests. Generally, the more severe the diagnosis, the faster the diagnostic and decision-making phase. However, irrespective of the actual timeline, time was experienced as ambivalent, slow, exhausting, unbearable, and like being "on a rollercoaster". However, time as well served as mental and emotional preparation for making and deciding to terminate.

*"Mum: No matter how much I read, I couldn't take it in […] and wouldn't take it in. No. But I really think it's been import-ant that it has been so long (the diagnostic process), for us at least. […] The experience (of giving birth) was much calmer than I think it would have been if I had to make the decision in a couple of days" (interview 1)*

## Discussion

This study investigated the diagnostic process and decision-making involved in terminating a pregnancy from the per-spective of expectant parents. Our findings highlight an two-phased, emotional rollercoaster of hope/no hope during this process and the difficulties of navigating medical and genetic information often experienced as uncertain or incomplete. Our findings demonstrate the complex factors that influence the TOPFA decision-making process. The couples highlighted the importance of decisional alignment between couples, but recognized that the woman's decision-making process could be longer due to the embodied nature of pregnancy.

### The worry and value of time

A prenatal diagnostic process is distressing for expectant parents [1,5,9,10,23,24]. Healthcare professionals and parents alike must navigate this ambiguous phase due to the time it takes to complete diagnostic procedures and monitor fetal development [1]. Even if a diagnosis is reached, prognosis can be difficult to predict, and this combination of waiting time and uncertainty may cause parental frustration. Several studies have highlighted the importance of continuous information [1,11,13], sensitive communication [2,24,25], and accessible healthcare professionals [5,13,26].

Our results identified two distinct phases in the diagnostic process: The first phase was relatively short, but the time from suspected findings to specialist evaluation was characterized by concern and frustration due to the experienced unwillingness of the sonographers to share information about their suspicion. While acknowledging the participant's frustrations and complicated feelings at this point, we maintain that the best clinical practice is to await specialist eval-uation before disclosing information in order to ensure accuracy and avoid misunderstandings and inaccuracies [6,24]. However, as this phase was experienced as particularly tense and difficult by participants, it may be useful for the first-line healthcare professionals to explicitly communicate this reasoning to expectant parents, as remaining silent may be more detrimental.

The specialist evaluation initiated a second, and often much longer, phase in the diagnostic process, where – in con-trast to the first phase – the participants described a sense of companionship with trusted healthcare professionals. It is

well known waiting times and delay may cause stress and anxiety in expectant parents [9,24,27], but our results suggest that feelings of companionship may alleviate the situation. This is supported by a recent review, showing that parents undergoing TOPFA value specialized care [25] and continuity [24]. Our results indicate that time to be informed, adjust and reflect in companionship with healthcare professionals can be valuable in reaching a value-consistent and informed decision with less doubt. Less doubt is desirable, as confidence in the decision has previously been correlated with less complicated grief after the termination [28,29].

Our results show that the conundrum of "time" during the diagnostic process encompasses unbearable worry but also space to reflect and adjust. Based on this finding, we suggest that healthcare professionals may more comfortably lean into an acceptance that time spent in this vacuum, even if burdensome, is needed for maturing a decision that is consistent with parents' individual hopes and concerns and that time in this vacuum may be manageable for parents when accompanied with an empathic specialist.

### Embodiment and alignment in decision making

Our results show how the TOPFA decision manifested after continuous contemplation of the pros and cons, hope and hopelessness, molding the decision along the way. Many participants had previously discussed and agreed on a hypothetical termination of pregnancy in the case of a severely affected fetus, but these hypothetical, pre-pregnancy decisions clashed with the complexity of reality, where the decision to undergo TOPFA was revisited, resisted, discussed and tested between the couple before reaching the final decision.

Our study included the fathers' experiences of a prenatal diagnostic process and the TOPFA decision, which is rare [30]. A central finding was the fathers' concern for alignment in decision-making, and ensuring that their pregnant partner was not rushed in the decision-making process, which has also been suggested elsewhere [10]. A few previous studies have focused on women's relation to the fetus/baby following a prenatal diagnosis and found contrasting feelings of stronger attachment and emotional distance [1,23,31]. This study adds additional perspectives by showing how the participants (male and female) acknowledged the bodily aspect of pregnancy as a factor in the process of decision-making, postponing the decision to due to the embodied pregnancy experience and feeling of being bodily engaged with the fetus/baby. In this study, the male partners described being able to withdraw from the fetus/baby, as they were not bodily attached, and they recognized how that created a different (temporal) context for decision-making process for the woman.

### Strengths and limitations

A strength of the present study was the inclusion of both woman and partner perspectives, as research on men's experiences with TOPFA is limited. Recruitment continued until data was sufficiently rich and varied to provide for a robust analysis. Future research could explore healthcare professionals' experiences of the prenatal diagnostic process and how to support expectant parents in making informed and value-consistent reproductive choices.

### Implications for practice

The study highlights the need for clinicians in prenatal diagnostics to recognize how prior reproductive experiences and emotional context shape women's interpretation of diagnostic information. When women express hesitation or uncertainty, these should be seen as opportunities for dialogue. An interpretive approach—exploring how patients understand their situation and what values guide them—can help rebuild trust, especially in cases of prior conflicting or delayed information. Such engagement supports clearer communication and more ethically grounded decision-making in situations marked by ambiguity or emotional distress.

### Conclusion

This qualitative study showed the intertwined characteristics of diagnostic processes and decision-making for parents undergoing TOPFA. The participants described a feeling of timelessness where hope and hopelessness continuously

replaced each other. The situation was described as unpredictable, lonely and intangible because the decision to terminate or continue the pregnancy rested solely on their shoulders. Participants highlighted agreement and awareness of different bodily attachment as prominent aspects in the decision. The process was two-phased, with a first phase of feeling alone and kept in the dark by the sonographer. The second phase was characterized by a feeling of companionship with healthcare professionals and was reported to be valuable in preparing for whether to terminate the pregnancy or not. Based on these findings, we suggest that the diagnostic process is a productive time where expectant parents may have the necessary time and space – even if uncomfortable – to grasp, prepare and shape their decision in an individualized and informative companionship with trusted healthcare professionals.

## Acknowledgments

The authors want to extend their sincerest gratitude to all the participants who took time to share their experiences with us. Also, much thanks to Caroline Margaret Moos, Department of Clinical Research, University Hospital of Southern Denmark, for careful proof reading and language editing.

## Author contributions

**Conceptualization:** Maiken Fabricius Damm, Dorte Hvidtjoern, Puk Sandager, Ida Vogel, Stina Lou.

**Data curation:** Maiken Fabricius Damm, Puk Sandager, Stina Lou.

**Formal analysis:** Maiken Fabricius Damm, Stina Lou.

**Funding acquisition:** Maiken Fabricius Damm, Dorte Hvidtjoern, Ida Vogel.

**Investigation:** Maiken Fabricius Damm.

**Methodology:** Maiken Fabricius Damm, Christina Prinds, Stina Lou.

**Project administration:** Maiken Fabricius Damm.

**Resources:** Dorte Hvidtjoern.

**Supervision:** Dorte Hvidtjoern, Puk Sandager, Ida Vogel, Christina Prinds, Stina Lou.

**Validation:** Dorte Hvidtjoern, Puk Sandager, Ida Vogel, Christina Prinds, Stina Lou.

**Writing – original draft:** Maiken Fabricius Damm.

**Writing – review & editing:** Maiken Fabricius Damm, Dorte Hvidtjoern, Puk Sandager, Ida Vogel, Christina Prinds, Stina Lou.

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
