## [Decision Letter · Decision Letter 0]

24 Apr 2025

PONE-D-25-06657Parental experience of the diagnostic process and its role in the decision to terminate pregnancy due to fetal abnormality: A qualitative interview studyPLOS ONE

Dear Dr. Lou,

Thank you for submitting your manuscript to PLOS ONE. After careful consideration, we feel that it has merit but does not fully meet PLOS ONE’s publication criteria as it currently stands. Therefore, we invite you to submit a revised version of the manuscript that addresses the points raised during the review process.

We look forward to receiving your revised manuscript.

Kind regards,

Patrick Ifeanyi Okonta, MBBCh, MPH, FWACS, FMCOG, MD, DRH

Academic Editor

PLOS ONE

 [This work was funded by University Hospital of Southern Denmark (grant#23) and by Aarhus University Hospital, (grant#SVP101121 and SVP040723).]. 

5. In the online submission form, you indicated that [For this study, participants only consented to external data sharing in anonymized form. Since full transcripts cannot be fully anonymized due to the highly individual context, the transcripts can only be made available upon reasonable request and special conditions may apply. Any requests concerning data access can be directed to defactum@rm.dk].

6.  Please include a caption for figure 1.

Reviewers' comments:

Reviewer's Responses to Questions

**Comments to the Author**

1. Is the manuscript technically sound, and do the data support the conclusions?

Reviewer #1: Yes

Reviewer #2: Yes

2. Has the statistical analysis been performed appropriately and rigorously? 

Reviewer #1: N/A

Reviewer #2: Yes

3. Have the authors made all data underlying the findings in their manuscript fully available?

Reviewer #1: Yes

Reviewer #2: Yes

4. Is the manuscript presented in an intelligible fashion and written in standard English?

Reviewer #1: Yes

Reviewer #2: Yes

5. Review Comments to the Author

Reviewer #1: 1) hermeneutical and phenomenological approach needs to be explained more.

2) As authors have used semi-structured interviews, it should be explained that how interview guide was developed?

3) Justify that how much time after abortion the interviews have been done?

4) Implications of study should be extended more.

5) Adding some more demographic and clinical characteristics of participants is worthy.

Reviewer #2: The manuscript is well written and recommended for it's technicality and it's originality. But I would suggest the author make a tabular thematic analysis between the two themes as stated by the author.

6. PLOS authors have the option to publish the peer review history of their article (what does this mean? ). If published, this will include your full peer review and any attached files.

**Do you want your identity to be public for this peer review?** For information about this choice, including consent withdrawal, please see our Privacy Policy .

Reviewer #1: **Yes: ** Prof. Mohammad Amin Bahrami

Reviewer #2: **Yes: ** Victor Eyo Assi (PhD)

---

## [Author Response · Author response to Decision Letter 1]

14 May 2025

I have uploaded a response letter

---

## [Editor Report · Decision Letter 1]

19 May 2025

Parental experience of the diagnostic process and its role in the decision to terminate pregnancy due to fetal abnormality; a qualitative interview study

PONE-D-25-06657R1

Dear Dr. Lou,

We’re pleased to inform you that your manuscript has been judged scientifically suitable for publication and will be formally accepted for publication once it meets all outstanding technical requirements.

Kind regards,

Patrick Ifeanyi Okonta, MBBCh, MPH, FWACS, FMCOG, MD, DRH

Academic Editor

PLOS ONE
---

## [Editor Report · Acceptance letter]

PONE-D-25-06657R1

PLOS ONE

Dear Dr. Lou,

I'm pleased to inform you that your manuscript has been deemed suitable for publication in PLOS ONE. Congratulations! Your manuscript is now being handed over to our production team.

Kind regards,

on behalf of

Professor Patrick Ifeanyi Okonta

Academic Editor

PLOS ONE